# Path sampling of recurrent neural networks by incorporating known physics

Sun-Ting Tsai[1,2], Eric Fields[3,4], Yijia Xu[1,2,5], En-Jui Kuo[1,5] & Pratyush Tiwary ◉[2,3] ✉

Recurrent neural networks have seen widespread use in modeling dynamical systems in varied domains such as weather prediction, text prediction and several others. Often one wishes to supplement the experimentally observed dynamics with prior knowledge or intuition about the system. While the recurrent nature of these networks allows them to model arbitrarily long memories in the time series used in training, it makes it harder to impose prior knowledge or intuition through generic constraints. In this work, we present a path sampling approach based on principle of Maximum Caliber that allows us to include generic thermodynamic or kinetic constraints into recurrent neural networks. We show the method here for a widely used type of recurrent neural network known as long short-term memory network in the context of supplementing time series collected from different application domains. These include classical Molecular Dynamics of a protein and Monte Carlo simulations of an open quantum system continuously losing photons to the environment and displaying Rabi oscillations. Our method can be easily generalized to other generative artificial intelligence models and to generic time series in different areas of physical and social sciences, where one wishes to supplement limited data with intuition or theory based corrections.

Artificial neural networks (ANNs) and modern-day Artificial Intelligence (AI) seek to mimic the considerable power of a biological brain to learn information from data and robustly perform a variety of tasks, such as text and image classifications, speech recognition, machine translation, and self-driving cars[1–9]. In recent years, ANNs have been shown to even outperform humans in certain tasks such as playing board games and weather prediction[10–12]. Closer to physical sciences, ANNs have been used to make predictions of folded structures of proteins[13], accelerate all-atom molecular dynamics (MD) simulations[14–17], learn better order parameters in complex molecular systems[18–21], and many other exciting applications. While the possible types of ANNs is huge, in this work we are interested in Recurrent Neural Networks (RNNs). These are a class of ANNs that incorporate memory in their architecture allowing them to directly capture temporal correlations in time series data[22,23]. Furthermore, RNN

frameworks such as long short-term memory (LSTM) neural networks[24] can account for arbitrary and unknown memory effects in the time series being studied. These features have made RNNs very popular for many applications such as weather, stock market prediction and dynamics of complex molecular systems[6,12,25,26]. In such applications, the assumption of independence between data points at different time steps is also invalid, and furthermore events that occurred at an arbitrary time in the past can have an effect on future events[6,12,23,26].

In spite of their staggering success, one concern applicable to RNNs and ANNs in general is that they are only able to capture the information present in their training datasets, unless additional knowledge or constraints are incorporated. Since a training dataset is limited by incomplete sampling of the unknown, high-dimensional distribution of interest, this can cause a model to overfit and not

[1]Department of Physics, University of Maryland, College Park, MD 20742, USA. [2]Institute for Physical Science and Technology, University of Maryland, College Park, MD 20742, USA. [3]Department of Chemistry and Biochemistry, University of Maryland, College Park, MD 20742, USA. [4]Department of Computer Science, University of Maryland, College Park, MD 20742, USA. [5]Joint Quantum Institute and Joint Center for Quantum Information and Computer Science, NIST/University of Maryland, College Park, MD 20742, USA. ✉e-mail: ptiwary@umd.edu

precisely represent the true distribution[27]. For instance, in the context of generalizing and extrapolating time-series observed from finite length simulations or experiments, partial sampling when generating a training dataset is almost unavoidable. This may come from only being able to simulate dynamics on a particular timescale that is not long enough to completely capture characteristics of interest[28] or simply thermal noise. This could then manifest as a misleading violation of detailed balance[29], and a RNN model trained on such a time-series would dutifully replicate these violations. In such cases, enforcing physics inspired constraints corresponding to the characteristics of interest when training an RNN-based model is critical for accurately modeling the true underlying distribution of data.

Given the importance of this problem, numerous approaches have been proposed in the recent past to add constraints to LSTMs, which we summarize in Sec. 2.2. However, they can generally only deal with very specific types of constraints, complicated further by the recurrent or feedback nature of the networks[30–32]. In this work we provide a generalizable, statistical physics based approach to add a variety of constraints to LSTMs. To achieve this, we use ideas of path sampling combined with LSTM, facilitated through the principle of Maximum Caliber. Our guiding principle is our previous work[26] where we show that training an LSTM model is akin to learning path probabilities of the underlying time series. This facilitates generating a large number of trajectories in a controlled manner and in parallel, that conform to the thermodynamic and dynamic features of the input trajectory. From these, we select a sub-sample of trajectories that are consistent with the desired static or dynamical knowledge. The bias due to sub-sampling is accounted for using the Maximum Caliber framework[33] by calculating weights for different possible trajectories. A new round of LSTM is then trained on these sub-sampled trajectories that in one-shot combines observed time series with known static and dynamical knowledge. This framework allows for constrained learning without incorporating an explicit constraint within the loss function.

We demonstrate the usefulness of our approach by adding thermodynamic and kinetic constraints to several problems, including a 3-state Markov model, a synthetic peptide $\alpha$-aminoisobutyric acid 9 (Aib9) in all-atom water and an open quantum system continuously dissipating photons to the environment. Irrespective of the origin of the dynamics, the approach developed here, which we call Path Sampling LSTM, is shown to be capable of blending prior physics based constraints with the observed dynamics in a seamless manner. Apart from its practical relevance, we believe the Path Sampling LSTM approach also provides a computationally efficient way of exploring the trajectory space of generic physical systems, and investigating how the thermodynamics of trajectories changes with different constraints[34].

## Results

### Recapitulating long short-term memory (LSTM) networks

In this work, we use long short-term memory (LSTM) networks[24,26] to generated trajectories conforming to prior knowledge. We first summarize what LSTMs are. These are a specific class of RNNs that work well for predicting time series through incorporating feedback connections whereby past predictions are used as input for future predictions[24]. We have shown in ref. 26 that we can let a simple language model built upon LSTM learn a generative model from time series generated from dynamical simulations or experiments performed upon molecular systems of arbitrary complexity. Such a time series used can be denoted as $\{\chi^{(t)}\}$, where $\chi \in \mathbb{R}$ is a one-dimensional order parameter corresponding to some collective properties of a higher-dimensional molecular system and $t$ denotes time. $\chi$ is discretized into $N$ states represented by a set of $N$-dimensional binary vectors or one-hot encoding vectors $\mathbf{v}^{(t)}$. These one-hot vectors have an entry equalling one for the representative state and all the other entries are set to zeros. In ref. 26, we also introduced the embedding

layer from language processing into the LSTM architecture to learn molecular trajectories. The embedding layer maps the one-hot vectors $\mathbf{v}^{(t)}$ to a $M$-dimensional densely distributed vector $\mathbf{x}^{(t)}$ via

$$\mathbf{x}^{(t)} = \Lambda \mathbf{v}^{(t)} \tag{1}$$

where $\Lambda$ is the embedding matrix and serves as a trainable look-up table. The time series of the dense vectors $\mathbf{x}^{(t)}$ is then used as the input for LSTM.

A unique aspect of LSTMs is the use of a gating mechanism for controlling the flow of information[35,36]. It uses $\mathbf{x}^{(t)}$ as input to generate a $L$-dimensional hidden vector $\mathbf{h}^{(t)}$, where $L$ is called the RNN or LSTM unit and $\mathbf{h}^{(t)}$ is called the hidden unit. The $\mathbf{x}^{(t+1)}$ and $\mathbf{h}^{(t)}$ are then used as the inputs for the next time step following the equations of forward propagation as described in the Supplementary Information (SI). The hidden state $\mathbf{h}^{(t)}$ at each time step is also mapped to a final output vector $\hat{\mathbf{y}}^{(t)}$ through a dense layer. This final output $\hat{\mathbf{y}}^{(t)}$ is then interpreted as the probability for any state to happen as obtained through minimizing the loss function $J$ defined below:

$$J = -\sum_{t=0}^{T-1} \mathbf{v}^{(t+1)} \cdot \ln \hat{\mathbf{y}}^{(t)} \tag{2}$$

where $T$ is the length of the input time series[26].

### Previous approaches to add constraints to LSTM networks and their limitations

A naive way of applying constraints when training LSTMs is incorporating a term within the loss function in Eq. (2) whose value decreases as the model's adherence to the constraint increases. This approach has been successfully applied for instance in the context of 4-D flight trajectory prediction[30]. A limitation of this naive approach is that the desired constraint must have an explicit mathematical formulation parameterized by the RNN's raw output, so that the value of the regularization term in the constraint can be adjusted through training. In the case of LSTMs, the raw output of the model passed through a softmax layer is equivalent to the probability of a future event conditioned on an observed past event. Formulating mathematical constraints solely in terms of such conditional probabilities has been done for specific constraints[30] and can be very challenging in general. Alternative more nuanced approaches to enforcing constraints in LSTMs have also been employed specific to the particular application. For example, when applying LSTMs to generate descriptions of input images, ref. 31 constrained part of speech patterns to match syntactically valid sentences by incorporating a part of speech tagger, that tags words as noun, verb etc. within a parallel LSTM language model architecture. The success of this approach relies on being able to reliably introduce more information to the model through the predictive part of the speech tagger. In applying LSTMs to estimating geomechanical logs, ref. 32 incorporated a physical constraint by adding an additional layer into the LSTM architecture to represent a known intermediate variable in physical models. The success of this approach as well relies on utilizing a known physical mechanism involved in the specific engineering problem.

### Our approach: path sampling LSTM

The approaches described in Section "Previous approaches to add constraints to LSTM networks and their limitations" while useful in the specific contexts for which they were developed, are not generally applicable to different constraints. For instance, when combining experimental time series for molecular systems with known theoretical knowledge, the constraints are often meaningful only in an ensemble-averaged sense. This per definition involves replicating many copies of the same system. With dynamical constraints involving rates of transitions, the problem is arguably even harder as it involves averaging

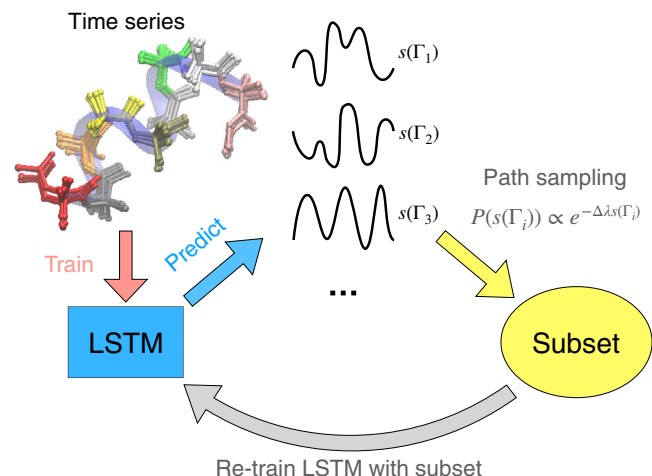

**Fig. 1 | Procedure for path sampling LSTM.** This schematic plot shows the workflow for constraining some static or dynamical variable $s(\Gamma_i)$, given an unconstrained LSTM model. The workflow begins with generating numerous predicted trajectories from the constraint-free LSTM model. The corresponding variables that we seek to constrain can be calculated from the predicted trajectories and are denoted by $s(\Gamma_1), s(\Gamma_2), s(\Gamma_3)$ in the plot. We then perform path sampling and select a smaller subset of trajectories in a biased manner that conforms to the desired constraints, with a probability $P(s(\Gamma_i)) \propto e^{-\Delta\lambda s(\Gamma_i)}$, where $\Delta\lambda$ is solved by the Eq. (6). The subset is then used as a new dataset to train the LSTM model.

over path ensembles. Our statistical physics based approach deals with these issues in a self-contained manner, facilitated by our previously derived connections between LSTM loss functions and path entropy[26].

Our key idea behind constraining recurrent neural networks with desired physical properties is to sample a subset from predicted trajectories generated from the trained LSTM models. The sampling is performed in a way such that the subset satisfies desired thermodynamic or dynamic constraints. For a long enough training set, we have shown in our previous work[26] that LSTM learns the path probability, and thus a trained LSTM generates copies of the trajectory from the correct path ensemble. In other words, the final output vector $\hat{\mathbf{y}}^{(t)}$ will learn how to generate $P_\Gamma \equiv P(\mathbf{x}^{(0)}...\mathbf{x}^{(T)})$, where $P_\Gamma$ is the path probability associated to a specific path $\Gamma$ in the path ensemble characterized by the input trajectory fed to the LSTM. The principle of Maximum Caliber or MaxCal[33,37,38] provides a way to build dynamical models that incorporate any known thermodynamic or dynamic i.e. path-dependent constraints into this ensemble. Per MaxCal[33], one can derive $P_\Gamma$ by maximizing the following functional called Caliber:

$$\mathcal{C} = \sum_\Gamma P_\Gamma \ln \frac{P_\Gamma}{P_\Gamma^U} - \sum_i \lambda_i \left( \sum_\Gamma s_i(\Gamma) P_\Gamma - \bar{s}_i \right) \quad (3)$$

where $\lambda_i$ is the Lagrange multiplier associated to the $i$-th constraint that helps enforce path-dependent static or dynamical variables $s_i(\Gamma)$ to desired path ensemble averaged values $\bar{s}_i$. With appropriate normalization conditions for probabilities, maximizing Caliber in Eq. (3) relates the constrained path probability $P_\Gamma^*$ to the reference or unconstrained path probability $P_\Gamma^U$ as follows:

$$P_\Gamma^* \propto e^{-\sum_i \lambda_i^* s_i(\Gamma)} P_\Gamma^U \quad (4)$$

where the similar derivation for continuous time can be found using variational principle in Ref. 38.

From Eq. (4), it is easy to show that for two dynamical systems labeled A and B that only differ in the ensemble averaged values for some $j$-th constraint being $\bar{s}_j^A$ and $\bar{s}_j^B$, then their respective path

probabilities for some path $\Gamma$ are connected through:

$$P_\Gamma^B \propto e^{-\Delta\lambda_j s_j(\Gamma)} P_\Gamma^A \quad (5)$$

where $\Delta\lambda_j = \lambda_j^B - \lambda_j^A$.

With this formalism at hand, we label our observed time series as the system A and its corresponding path probability as $P_\Gamma^A$. This time series or trajectory has some thermodynamic or dynamical $j$-th observable equaling $\bar{s}_j^A$. On the basis of some other knowledge coming from theory, experiments or intuition, we seek this observable to instead equal $\bar{s}_j^A$. In accordance with ref. 26 we first train a LSTM that learns $P_\Gamma^A$. Our objective now is to train a LSTM model that can generate paths with probability $P_\Gamma^B$ with desired, corrected value of the constraint. For this we use Eq. (5) to calculate $\Delta\lambda$. This is implemented through the following efficient numerical scheme. We write down the following set of equations:

$$\begin{aligned} \bar{s}_j^B &= \sum_\Gamma P_\Gamma^B s_j(\Gamma) \\ &= \frac{\sum_{k \in \Omega} s_j(\Gamma_k) e^{-\Delta\lambda_j s_j(\Gamma_k)}}{\sum_{k \in \Omega} e^{-\Delta\lambda_j s_j(\Gamma_k)}} \end{aligned} \quad (6)$$

where $\Omega$ is the set of labeled paths sampled from the path probability $P_\Gamma^A$. By solving for $\Delta\lambda_j$ from Eq. (6) we have the sought $P_\Gamma^B$. In practice, this is achieved through the procedure depicted in Fig. 1, where the LSTM model trained with time series for the first physical system is used to generate a collection of predicted paths with a distribution proportional to path probability $P_\Gamma^A$. A re-sampling with an appropriate estimate of $\Delta\lambda_j$ is then performed to build a subset. This value is obtained by computing the right hand side of the second line in Eq. (6) over the resampled subset such that correct desired value of the constraint is obtained. This subset denotes sampling from the desired path probability $P_\Gamma^B$ and is used to re-train a new LSTM that will now give desired $\bar{s}_j^B$. The method can be easily generalized to two or more constraints. For example, in order to solve for two constraints, we can rewrite Eq. (5) as

$$P_\Gamma^B \propto e^{-\Delta\lambda_j s_j(\Gamma) - \Delta\lambda_k s_k(\Gamma)} P_\Gamma^A \quad (7)$$

where $\Delta\lambda_j$ and $\Delta\lambda_k$ are two unknown variables to be solved with two equations for the ensemble averages $\bar{s}_j^B$ and $\bar{s}_k^B$.

Henceforth, we refer to the unconstrained version of LSTM as simply LSTM and the constrained version introduced here as ps-LSTM for "path sampled" LSTM. Note that ps-LSTM is only used to avoid confusion with the unconstrained LSTM. Therefore, we do not change the actual infrastructure, mathematics of the LSTM procedure, or even claim that ps-LSTM is a new variant of LSTM.

## Numerical examples
In what follows, we will provide illustrative examples to elaborate the protocol developed in Section "Our approach: Path sampling LSTM". Here we show how we can constrain static and dynamical properties for different time series. Without loss of generality, we restrict ourselves to time series obtained from MD and Monte Carlo (MC) simulations of different classical and quantum systems. The protocol should naturally be applicable to time series from experiments such as single molecule force spectroscopy[26]. These include (1) time series of a model 3-state system following Markovian dynamics, and (2) non-Markovian dynamics for the synthetic peptide Aib9 undergoing conformational transitions in all-atom water. The non-Markovianity for Aib9 arises because we project the dynamics of the 14,241-dimensional system onto a single degree of freedom. Our neural networks were built using Pytorch 1.10[39]. Further details of system/neural network parametrization as well as training and validation details are provided in the SI.

## Three-state Markovian dynamics

For the first illustrative example, we apply LSTM to a 3 state model system following Markovian dynamics for moving between the 3 states. This system, comprising states labeled 0, 1 and 2 is illustrated in Fig. 2(a). Figure 2(a) also shows the state-to-state transition rates for the unconstrained system. We then seek to constrain the average number of transitions per unit time between states 0,1 and 1,2 as defined below

$$\langle N \rangle = \frac{1}{L_{\text{traj}}}(N_{0\leftrightarrow1} + N_{1\leftrightarrow2}) \tag{8}$$

where $L_{\text{traj}}$ is the length of trajectory and $N_{0\leftrightarrow1}$ and $N_{1\leftrightarrow2}$ are the number of times a transition occurs between states 0 and 1 or states 1 and 2 respectively. This example can then be directly compared with the analytical result Eq. (18) derived in the Appendix, thereby validating the findings from ps-LSTM.

Given the transition kernel shown in Fig. 2 (a), we generate a time series that conforms to it. Following Sec. 2.3, we train ps-LSTM using this time series and the constraint on $\langle N \rangle$ described in Eq. (8). As per the Markovian transition kernel we have $\langle N \rangle = 0.0894$, while we seek to constrain it to 0.13. In other words, given a time series we want to increase the number of transitions per unit time between 2 of the 3 pairs of states. In Fig. 2 (b), we show the transition kernel obtained from the time series generated by ps-LSTM via direct counting. In particular, we would like to highlight that when enforcing a faster rate of state-to-state transitions sampling to increase the average number of nearest neighbor transitions, the transition matrix of ps-LSTM predictions show correspondingly increased rates of transition without completely destroying the original kinetics of the system. Using Eq. (19) provided in Appendix, we can predict the new transition kernel given by ps-LSTM. The comparison is also shown in SI.

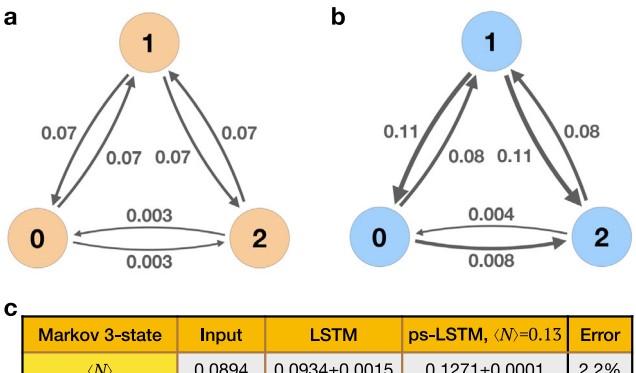

| Markov 3-state | Input | LSTM | ps-LSTM, ⟨N⟩=0.13 | Error |
|---|---|---|---|---|
| $\langle N \rangle$ | 0.0894 | 0.0934±0.0015 | 0.1271±0.0001 | 2.2% |

**Fig. 2 | 3 state Markovian system: LSTM, ps-LSTM and analytical predictions.** Here we show results of applying ps-LSTM to the 3 state Markovian system where we constrain $\langle N \rangle$. In (**a**), we provide the input transition kernel without constraints. In (**b**), we show the transition kernel obtained from ps-LSTM generated time-series via direct counting, where we achieve a $\langle N \rangle$ close to the target $\langle N \rangle$=0.13. The calculated values for $\langle N \rangle$ are shown in (**c**) for LSTM as the average of 100 predictions and for ps-LSTM as the average of 200 predictions. The error represents "error percentage" which is defined as the difference between ps-LSTM result and target value $\langle N \rangle$=0.13 divided by the target value.

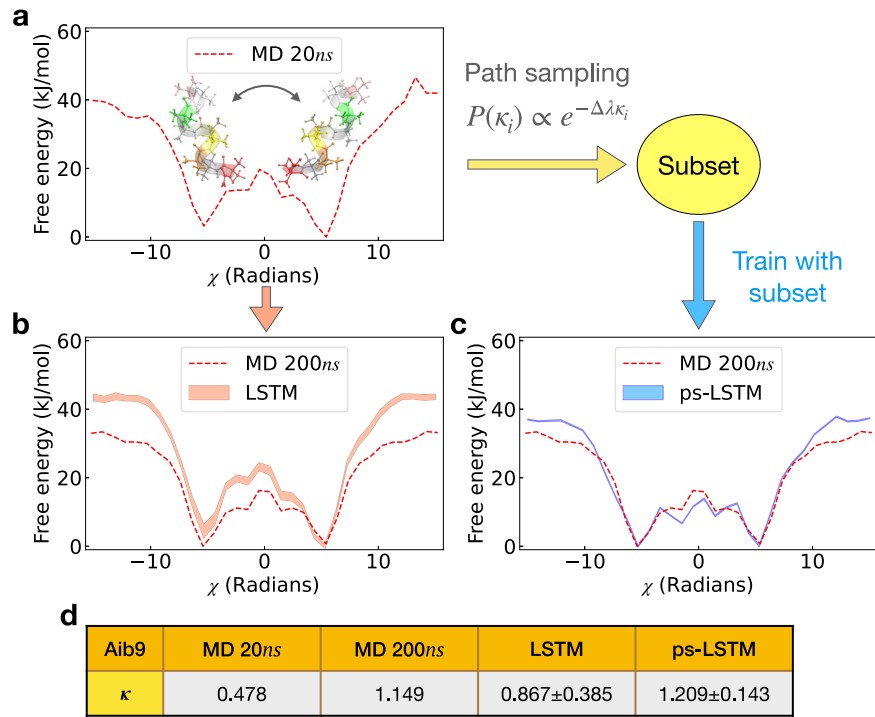

| Aib9 | MD 20*ns* | MD 200*ns* | LSTM | ps-LSTM |
|---|---|---|---|---|
| $\kappa$ | 0.478 | 1.149 | 0.867±0.385 | 1.209±0.143 |

**Fig. 3 | Comparing predictions at 200*ns* for different values of the symmetry parameter $\kappa$.** Here we show that ps-LSTM learns the correct symmetry $\kappa$. The original training data is a 20*ns* Aib9 trajectory generated from MD simulation at 500K, where (**a**) shows its calculated free energy profile has an asymmetry of population between L and R helix states. The snapshots of L and R configurations at $\chi$ = 5.2 and $\chi$ = − 5.31 are also displayed as insets above the free energy profile. Training LSTM model with this asymmetric data and using it to predict what would happen at 200*ns* leads to the result shown in (**b**), where the LSTM predictions retain and even enhance the undesired free energy asymmetry while the free energies calculated from a longer 200*ns* trajectory shows the desired symmetric profile. In (**c**), we show that ps-LSTM trained as described in Section "Equilibrium constraint on Aib9" can not only predict the correct symmetry, but also deviate less from the true free energy calculated from the reference 200*ns* data. The table in (**d**) shows the $\kappa$ values defined in Eq. (9) for different trajectories. The free energy profiles and the $\kappa$ values in (**b**) and (**c**) are averaged over 10 independent training processes. The corresponding error bars are calculated as standard errors and filled with transparent colors.

## MD simulations of $\alpha$-aminoisobutyric acid 9 (Aib9)

For our second, more ambitious application, we study the 9-residue synthetic peptide $\alpha$-aminoisobutyric acid 9 (Aib9)[40,41]. Aib9 undergoes transitions between fully left-handed (L) helix and fully right-handed (R) helix forms. This is a highly collective transition involving concerted movement of all 9 residues. During this global transition, there are many alternate pathways that can be taken, connected through a network of several lowly-populated intermediate states[40,41]. This makes it hard to find a good low-dimensional coordinate along which the dynamics can be projected without significant memory effects[40,41]. The problem is further accentuated by the presence of numerous high-energy barriers between the metastable states that result in their poor sampling when studied through all-atom MD. For example, through experimental measurements[42] and enhanced sampling simulations[40,41], the achiral peptide should show the same equilibrium likelihood of existing in the L and R forms. However, due to force-field inaccuracies[40] and insufficient sampling, MD simulations typically are too short to obtain such a result. In the first type of constraint, which enforces static or equilibrium probabilities, we show how our ps-LSTM approach can correct the time series obtained from such a MD simulation to enforce the symmetric helicity. In a second type of dynamical constraint, we show how we can enforce a desired local transition rate between different protein conformations.

### Equilibrium constraint on Aib9

We first discuss results for enforcing the constraint of symmetric helicity on Aib9, shown in Fig. 3. Here we have defined the free energy $F = -k_B T \ln P$, where $k_B$ and $T$ are the Boltzmann constant and temperature, and $P$ is the equilibrium probability calculated by direct counting from a respective time series. In Fig. 3 (a)–(c) we have projected free energies from different methods along the summation $\chi$ of the 5 inner dihedral angles $\phi$, which allows us to distinguish the L and R helices. We define $\chi \equiv \sum_{i=3}^{7} \phi_i$ and note that $\chi \approx 5.4$ and $\chi \approx -5.4$ for L and R respectively[41]. In order to have a reference to be compared with, we perform the simulation at temperature 500K under ambient pressure. As can be seen from Fig. 3(b), we are able to see a symmetric free energy profile after 100$ns$.

For LSTM to process the time series for $\chi$ as done in Ref. 26, we first spatially discretize $\chi$ into 32 labels or bins. To quantify the symmetry between left- and right-handed populations, we define a symmetry parameter $\kappa$:

$$\kappa = \frac{\sum_{i=0}^{i=15} P_i}{\sum_{i=16}^{i=32} P_i} \qquad (9)$$

where $_{Pi}$ denotes equilibrium probability for being found in bin label $i$. For symmetric populations we expect $\kappa \approx 1$. In Fig. 3 (a), we show the free energy from the first 20$ns$ segment of time series from MD. This 20$ns$ time series is then later used to train our LSTM model. It can be seen that the insufficient amount of sampling results in an incorrect asymmetry of populations between L and R helix states with $\kappa \approx 0.5$. We first train a constraint-free LSTM on this trajectory following Ref. 26 with which we generate a 200$ns$ time series for $\chi$. Figure 3(b) shows how a longer 200$ns$ MD trajectory would have been sufficient to converge to a symmetric free energy with $\kappa \approx 1$. However, Fig. 3(b) also shows the population along $\chi$ measured from the LSTM generated time series, which preserves the initially asymmetry that it witnessed in the original training trajectory.

In Fig. 3(c) we show the results from using ps-LSTM where we apply the constraint $\kappa = 1$. For this, we let the constraint-free LSTM model generate 200 indepedndent time series of length 20$ns$ long and used the method from Section "Our approach: Path sampling LSTM" 2.3 to enforce the constraint $\kappa = 1$ for 200$ns$ long time series. We calculate $\kappa$ values from the different predicted time series and use Eq. (6) to solve for an appropriate $\Delta\lambda$ needed for $\langle\kappa\rangle = 1$. We then

perform path sampling with a biased probability $\propto e^{-\Delta\lambda}$ to select 10 trajectories from the 200 predictions. These 10 time series were then used to construct a subset and train a new ps-LSTM. As can be seen in Fig. 3(c), ps-LSTM captures the correct symmetric free energy profile giving $\kappa = 1$. Interestingly, ps-LSTM also significantly reduces the deviations from the reference free energy at $|\chi| > 10$. In the SI, we have also provided the eigenspectrum of the transition matrix and shown that relative to LSTM, ps-LSTM pushes the kinetics for events across timescales in the correct direction. In Fig. 3(d), we show the $\kappa$ calculated from the trajectories of 20$ns$ and 200$ns$ MD simulations of Aib9 and from the predicted 200$ns$ trajectories of LSTM and ps-LSTM.

### Dynamical constraint on Aib9

Our second test is performed to enforce a dynamical constraint i.e. one that explicitly depends on the kinetics of the system[43]. Specifically, we constrain the ensemble averaged number of nearest neighbor transitions per unit time $\langle N \rangle$ along the sum of dihedral angle $\chi$ introduced in Section "Equilibrium constraint on Aib9". $\langle N \rangle$ is defined as

$$\langle N \rangle = \frac{1}{L_{\text{traj}}} \sum_i N_{i,i+1} \qquad (10)$$

where $L_{\text{traj}}$ is the length" of trajectory, and $N_{i,i+1}$ equals 1 if the values of $\chi$ at times $i$ and $i+1$ are separated only by a single bin, otherwise 0. The nearest neighbor transitions can be seen as a quantification of diffusivity when comparing the form of transition rate matrix from the discretized Smoluchowski equation to the one derived from principle of Maximum Caliber[43]. In Fig. 4(a), we show a free energy profile calculated from a 100$ns$ MD trajectory. As can be seen here, this trajectory is long enough to give symmetric populations for the L and R helix states. We find that the averaged number of nearest neighbor transitions $\langle N \rangle$ for this trajectory is approximately 0.4. In Fig. 4(a) we have also shown the free energy from a 200$ns$ long MD simulation which we use later for comparison. In Fig. 4(b), we show trajectory generated from training constraint-free LSTM[26] which follows the same Boltzmann statistics and kinetics as the input trajectory.

In order to constrain $\langle N \rangle$, we generate 800 independent time series from the constraint-free LSTM, and sample a subset consisting of 10 time series. With an appropriate $\Delta\lambda$, our path-sampled subsets are constrained to two different $\langle N \rangle$ values and used for training two distinct ps-LSTMs. In Fig. 4(c) and (d), we have shown the free energy profiles corresponding to ps-LSTM predictions trained on subsets with $\langle N \rangle = 0.38$ and $\langle N \rangle = 0.42$. As can be seen, compared to the actual 200$ns$ MD simulation of Aib9, the potential wells of L and R helix become narrower for $\langle N \rangle = 0.38$ and wider for $\langle N \rangle = 0.42$, which is the direct effect of changing fluctuations via nearest-neighbor transitions. Moreover, the potential barriers along $\chi$ become higher for $\langle N \rangle = 0.38$ and become lower for $\langle N \rangle = 0.42$. In Fig. 4(e), we provide the averaged transition times $\tau$ from L to R helix states and vice versa, where we can also see that the transition times do become longer for smaller $\langle N \rangle$ and shorter for larger $\langle N \rangle$, which is the expected result for decreased and increased diffusivities respectively[43,44].

To summarize so far, the results from constraining $\langle N \rangle$ show that through the path sampling method, ps-LSTM extrapolates the phenomena affected by changing small fluctuations, which was not provided in the training data set.

### Open quantum system

In this sub-section, we will demonstrate ps-LSTM applied to an open quantum system consisting of a single two-level atom with a single-mode cavity initially populated by seven photons, where the photons continuously dissipate to the environment via a certain dissipation rate (see Fig. 5). Simulating the dynamics of general open quantum systems has been a long-lasting challenge. As a result, several methods such as quantum jump approach and associated path sampling techniques

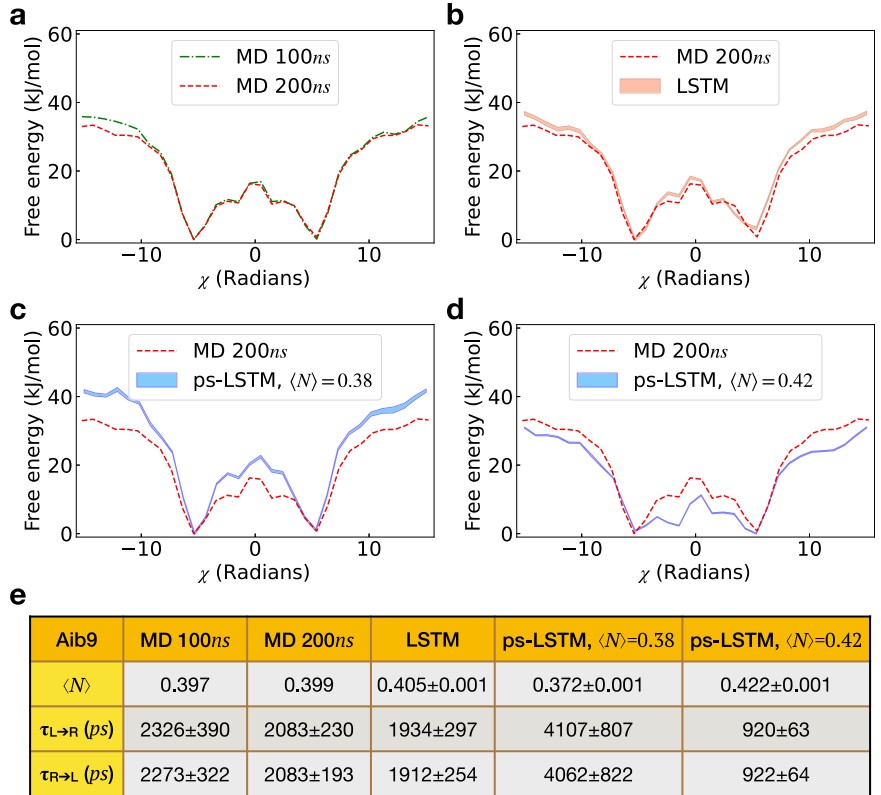

**Fig. 4 | Comparing predictions at 200ns for different values of the dynamical constraint ⟨N⟩.** In this plot, we show the free energy profiles calculated from (**a**) the 100ns trajectory in the training set, (**b**) both the actual 200ns trajectory and direct prediction from LSTM, (**c**) the reference 200ns trajectory and ps-LSTM prediction with constraint of nearest-neighbor (NN) transitions ⟨N⟩ = 0.38, and (**d**) the reference 200ns trajectory and prediction with constraint ⟨N⟩ = 0.42. The table in (**e**) lists the kinetic constraint ⟨N⟩ calculated from corresponding trajectories. The averaged transition time $\tau_{R \to L}$ and $\tau_{L \to R}$ in picoseconds were calculated by counting the numbers of transitions in each trajectory. For reference MD, the error bars were calculated by averaging over transition time in a single 100ns or 200ns trajectory, while for the predictions from LSTM and ps-LSTM, the error bars were averaged over 10 independent predictions with the transition time for each predicted trajectory calculated in the same way as MD trajectories. The free energy profiles and the first NN values ⟨N⟩ in (**b**)–(**d**) are averaged over 10 independent training processes. The corresponding error bars are calculated as standard errors and filled with transparent colors.

have been proposed[45–47]. Here, we combine the quantum jump approach[48,49] and ps-LSTM to generate quantum trajectories that provide correct expectation values of generic observables.

The example we study here is intrinsically hard because the system has a Hilbert space with at least 16 dimensions yet we only let LSTM see the individual quantum trajectories of a 1-dimensional observable. Although the quantum trajectories produced by Monte Carlo simulation in the full Hilbert space are Markovian, the dimensionality reduction from high-dimensional Hilbert space to the observable results in non-Markovian trajectories. In this example, we will let LSTM learn a dissipative observable which is the number of photons in the system. We will then show how we can use our ps-LSTM method to learn to predict trajectories with dissipation rate $\gamma = 0.2$ given training data consisting of only trajectories generated from simulations with $\gamma = 0.1$.

The time-evolution of an open quantum system with dim $\mathcal{H} = N$ is governed by Lindblad Master equation[50–53]:

$$\dot{\rho} = -\frac{i}{\hbar}[H, \rho] + \sum_{i=1}^{N^2-1} \gamma_i \left( L_i \rho L_i^\dagger - \frac{1}{2}\left\{ L_i^\dagger L_i, \rho \right\} \right). \quad (11)$$

where $\rho$ is the density matrix, $H$ is a Hamiltonian of the system, and $L_i$ are commonly called the Lindblad or jump operators of the system. $\gamma_i$ is the dissipation rate corresponding to jump operator $L_i$. For convenience, we choose the natural unit where $\hbar = 1$. For a large system, directly solving (11) is a formidable task. Therefore, an alternative approach is to perform Monte Carlo (MC) quantum-jump method[54–56],

which requires us to generate a large enough number of trajectories to produce correct expectation values of observables. Our training data for LSTM is therefore a set of quantum jump trajectories generated by the Monte Carlo quantum jump algorithm. Here we consider a simple two-level atom coupled to a leaky single-mode cavity through a dipole-type interaction[57]:

$$H_{\text{sys}} = \omega_1 a^\dagger a + \omega_2 \sigma_+ \sigma_- + g(\sigma_- a^\dagger + a\sigma_+) \quad (12)$$

where the $a$, $a^\dagger$ and $\sigma_-$, $\sigma_+$ are the annihilation and creation operators of photon and spin, respectively. Suppose above system is surrounded in a dissipative system which induces single-photon loss of cavity. In the quantum jump picture, we can write down following non-Hermitian Hamiltonian

$$H = H_{\text{sys}} - \frac{i\gamma}{2}a^\dagger a \quad (13)$$

where there is only one dissipation channel which is called photon emission with jump operator $\sqrt{\gamma}a$.

We use the built-in Monte Carlo solver in the QuTiP package[48,49] with a pre-selected dissipation rate $\gamma$ to generate a bunch of quantum jump trajectories of the cavity photon number $n_t$. It is important to note that although the Lindbladian and quantum jump method are Markov processes in Hilbert space, the quantum jump trajectories of $n_t$ learned by LSTM do not need to be Markovian in a coarse-grained state space ⟨n⟩.

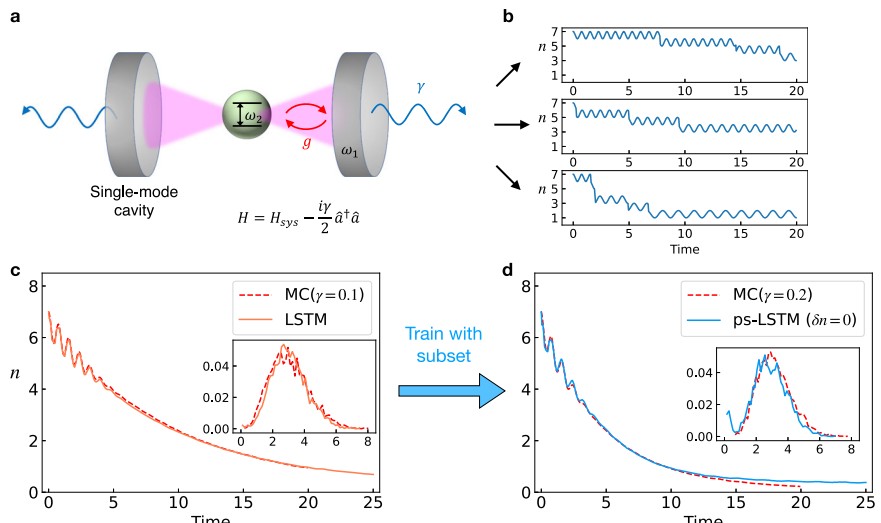

**Fig. 5 | Quantum jump trajectories generated from ps-LSTM. a** Schematic of the open quantum system we simulate. The system consists of a two-level atom surrounded by the cavity, where the initial state is chosen to be $|7\rangle \otimes |\uparrow\rangle$. The cavity photons not only experience interaction with the atom but also interact with the environment via continuously dissipating photons to the environment. The system can be described by the Hamiltonian written below the plot, where system Hamiltonian $H_{sys}$ is just Eq. (12) with $\omega_1 = \omega_2 = 2\pi$. **b** Some representative trajectories from the quantum jump simulations which we use to train LSTM and ps-LSTM. **c** The expectation value of number of photons $\langle n \rangle$ as a function of time obtained by averaging over 2000 MC simulations with $\gamma = 0.1$ (dashed red line) and 2000 predictions generated by LSTM (solid orange line). The inserted panel shows the distribution of the variance calculated over each trajectory, where the x-axis is the variance and the y-axis is the corresponding probability. **d** The expectation value of number of photons $\langle n \rangle$ as a function of time obtained by averaging over 2000 MC simulations with $\gamma = 0.2$ (dashed red line) and 2000 predictions generated by ps-LSTM (solid blue line). The inserted panel shows the distribution of the variance calculated over each trajectory.

In an approximate sense, the dissipation rate $\gamma$ appears as a parameter controlling the classically exponential decay of $\langle n_t \rangle$:

$$\langle n_t \rangle^{\text{theory}} \approx n_0 e^{-\gamma t} \qquad (14)$$

therefore, given the values of $\gamma$ and $t$, we can estimate the corresponding $\langle n_t \rangle^{\text{theory}}$. This $\langle n_t \rangle^{\text{theory}}$ will later be used as the constraint variable for ps-LSTM.

In general, the Lindbladian equation describes the time-evolution of a $N \times N$ matrix which is computationally challenging. However, the averaged trajectory of the observables, i.e. $\langle n_t \rangle$, is typically governed by a set of differential equations whose number of coefficients is much less than $N^2$. Previous work[58] has already demonstrated that standard LSTM can learn the feature of decaying pattern from the averaged trajectory $\langle n_t \rangle$, while it is definitely more useful yet challenging for the LSTM to learn the probabilistic model from the individual quantum trajectories $n_t$ and generate the stochastic trajectories with the correct expectation of the observable at every single time step since learning such stochastic trajectories allows us to do ps-LSTM and generate trajectories with a different dissipation rate.

Here we demonstrate how to apply ps-LSTM trained by individual trajectories from one dissipation rate to generate quantum trajectories with another dissipation rate. The parameters of Hamiltonian Eq. (12) are $\omega_1 = \omega_2 = 2\pi$, and $g = \frac{\pi}{2}$. As what we did in the previous example, we first spatially discretize $n_t$, which is the trajectories generated from the actual Monte Carlo quantum jump simulations with $\gamma = 0.1$, into 20 bins. We then let LSTM learn such trajectories and generate a set of predictions given only the starting condition of $n_t = 7$, as shown in Fig. 5(c). In Fig. 5(d), it can be seen that these predictions from LSTM follow the correct evolution curve averaged from the actual Monte Carlo quantum jump simulation with $\gamma = 0.1$. Next we constrain the LSTM model to learn a different dissipation rate $\gamma = 0.2$. In order to use ps-LSTM to sample $\gamma = 0.2$, we use Eq. (14) to estimate the corresponding $\langle n \rangle_t^*$ within the time interval $t \in (5, 7)$. Following the similar

spirit of s-ensemble[34,59], we define a dynamical variable $\delta n$, where

$$\delta n = \frac{1}{\Delta t} \sum_{j=1}^{K} \sum_{s}^{t + \Delta t} \| n_s^j - \langle n_s \rangle^{\text{theory}} \|^2 \qquad (15)$$

where $\langle n \rangle^{\text{theory}}$ is calculated from Eq. (14) with $\gamma = 0.2$. $K$ is the number of subsamples, which was chosen to be 2000. $t = 5$ and $\Delta t = 2$ are chosen such that minimizing $\delta n$ leads to a curve fit of exponential decay in classical regime. The ps-LSTM is then performed by constraining $\delta n = 0$. Constraining LSTM to learn a different $\gamma$ is very challenging if we only let LSTM learn the averaged trajectory as in Ref. 58, since the oscillating feature within the first 5 time units is a quantum mechanical effect and is hard to capture by simply changing $\gamma$.

However, by performing path sampling, we show that by constraining only the $\delta n$ in classical regime, ps-LSTM produced the correct quantum dynamics it captures from the quantum jump trajectories, which can be seen in Fig. 5(d). It is also worth noting that we actually perform a more challenging task in the prediction, where we let LSTM and ps-LSTM predict 5 time units more than the trajectories in the training set. That said, LSTM and ps-LSTM still give the prediction of $\langle n_t \rangle$ for $t > 20$, wherein it captures that the cavity photon number has been mostly dissipated and the averaged photon number does not change.

## Discussion

In this work, we proposed a method integrating statistical mechanics with machine learning in order to add arbitrary knowledge in the form of constraints to the widely used long short-term memory (LSTM) used for predicting generic time series in diverse problems from biological and quantum physics. These models are trained on available time series for the system at hand, which often have errors of different kinds. These errors could arise from either poor sampling due to rareness of the underlying events, or simply represent instrumentation errors. Using high fidelity artificial intelligence tools[26,60,61] to generate computationally cheaper copies of such time series is then prone to preserving such errors. Thus, it is extremely important to introduce systematic constraints that introduce prior knowledge in the LSTM network used to

replicate the time series provided in training. The recurrent nature of the LSTM and the non-Markovianity of the time series make it hard to impose such constraints in a trainable manner. For this, here our approach involves path sampling method with the principle of Maximum Caliber, and is called ps-LSTM. We demonstrated its usefulness on illustrative examples with varying difficulty levels and knowledge that is thermodynamic or kinetic in nature. Finally, as our method relies only on data post-processing and pre-processing, it should be easily generalized to other neural network models such as transformers and others[60,62], and for modeling time series from arbitrary experiments. In principle, our method can also be applied to non-physical systems as long as the distribution of path probability is known. We would also like to emphasize that equations such as Eq. (4) which are the cornerstone of this work, have been proposed previously in the literature for instance in the context of the dynamics of glass-forming liquids and the glass transition[34]. However, trajectory space is not easy to deal with, and our ps-LSTM approach provides a practical way of navigating this space.

## Methods

In this section, we develop useful, exact results for constraining state-to-state transitions in Markov processes that serve as useful benchmarking. It has been shown that if constraining pairwise statistics, maximizing Eq. (3) with appropriate normalization conditions yields the Markov process[63]

$$P_\Gamma^* = p_{i_0} \prod_{k=0}^{T-1} p_{i_k i_{k+1}} \tag{16}$$

where $p_{i_k i_{k+1}}$ are the time-independent transition probabilities defined by the Markov transition matrix. For such simple Markovian dynamics, we can easily solve for the outcome transition kernel by the $\Delta\lambda$ chosen.

Now we suppose we would like to adjust the frequency of transition from state $m$ to state $n$. With Eq. (16), following Ref. 63, we can rewrite Eq. (5) as

$$\prod_{k=0}^{T-1} p_{i_k i_{k+1}}^B \propto e^{-\Delta\lambda \sum_{k=0}^{T-1} \delta_{i_k,m} \delta_{i_{k+1},n}} \prod_{k=0}^{T-1} p_{i_k i_{k+1}}^A \tag{17}$$

where $\delta_{ij}$ is the Kronecker delta, equalling 1 when $i=j$ and 0 otherwise. Therefore, it can then be shown that

$$p_{mn}^B \propto e^{-\Delta\lambda} \cdot p_{mn}^A \tag{18}$$

Eq. (18) with predetermined $\Delta\lambda$ can be used to predict our numerical results.

Based on Eq. (8) and the equations in the Appendix, we can analyze the difference in transition kernel:

$$p_{mn}^{ps-LSTM} \propto e^{-\frac{\Delta\lambda}{L_{traj}}(\delta_{m0}\delta_{n1} + \delta_{m1}\delta_{n0} + \delta_{m1}\delta_{n2} + \delta_{m2}\delta_{n1})} \cdot p_{mn}^{LSTM} \tag{19}$$

where $\Delta\lambda$ we used is $-56.1$.

## Data availability

All data used in this study are available through github.com/tiwarylab/ps-LSTM.

## Code availability

All code associated with this work is available at github.com/tiwarylab/ps-LSTM.

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

## Acknowledgements

Research reported in this publication was supported by the National Institute of General Medical Sciences of the National Institutes of Health under Award Number R35GM142719. The content is solely the responsibility of the authors and does not necessarily represent the official views of the National Institutes of Health. The authors thank Deep-thought2, MARCC, and XSEDE (projects CHE180007P and CHE180027P) for providing computational resources used in this work. En-Jui Kuo and Yijia Xu are supported by ARO W911NF-15-1-0397, National Science Foundation QLCI grant OMA-2120757, AFOSR-MURI FA9550-19-1-0399, Department of Energy QSA program. The authors thank Yihang Wang, Dedi Wang, Yixu Wang, Jerry Yao-Chieh Hu, and Huan-Kuang Wu for discussions. We also thank Shams Mehdi for providing simulation trajectories and the analysis of the dihedral angles and Eric Beyerle for proofreading the manuscript. The LSTM and corresponding path sampling analysis code are available at github.com/tiwarylab.

## Author contributions

All authors designed research; P.T., S.T. and E.F. analyzed the 3-state system; P.T. and S.T. analyzed the Aib9 data; Y.X., E.K. and S.T. analyzed the open quantum system; S.T., E.F. and P.T. wrote the paper.

## Competing interests

The authors declare no competing interests.
