## [Peer Review File · Nature Communications]

REVIEWER COMMENTS

Reviewer #2 (Remarks to the Author):

The authors have proposed a novel application and instantiation for long short-term memory neural networks to mirror path-dependent constraints. The work is interesting, well-done, and novel; however, the writing and presentation of the paper make it somewhat difficult to follow and the authors should revise the article to make it more approachable for the likely general audience of Nature Communications.

In revisions, the authors can strengthen this paper by contextualizing the need for including path constraints in the LSTM model in the introduction (rather than focusing on the recent accomplishments of NNs), clarifying their procedure, and restructuring their figures to present their results more clearly (I have included some suggestions below).

According to the text, the authors train an LSTM to predict a path probability P_{Γ^A} , which they then use to subselect a set of paths P_{Γ^B} , using the principle of Maximum Caliber such that equation (6) corresponds to the target observable s_j^B . From here, they train a new LSTM to reconstruct their P_{Γ^B} , from which they can obtain the target observable at longer time scales.

Firstly, I'm hesitant to call this a variant of an LSTM (i.e., using a new name of ps-LSTM) or "incorporating" these constraints into the NN, as, in my reading, the authors do not change the actual infrastructure or mathematics of the LSTM procedure, but rather use the Maximum Caliber principle to do data sub-selection. For example, it's not necessarily fair to compare results obtained from an unconstrained and a constrained LSTM -- they are both using the same LSTM procedure, but with different training data. As it is more reflective of the procedure, the authors should rename "ps-LSTM" to "LSTM+PS" to denote the combination of the two methods, rather than an augmentation of LSTM.

Concerning sub-selection, what is the typical percentage of paths retained? One could argue that the number of simulations needed to approximate P_{Γ^A} should be pre-emptively increased considering the reduction to the subset -- how did the authors ensure that their training sizes were sufficient in each LSTM model? Note that this is a different question from asking whether the training sizes were sufficient to distinguish P_{Γ^A} and P_{Γ^B} , as the authors cover this nicely in SI VI.

Given that the training procedure includes two approximations: 1) approximating a subset of paths along the target path probability and 2) using this subset to extrapolate longer trajectories, where do the authors attribute their sources of error? It would be good to include an estimate of s_j^B for the intermediary subset in the main text to understand where the subset sits with respect to the initial and target path distributions.

The authors should restructure their figures to report the errors with respect to the target value rather than the unconstrained LSTM procedure (as discussed above). For example, in Figure 2, LSTM+PS achieves a value of 0.1271 ± 0.0001 with a target of 0.13; however, in the current table, it could appear that the LSTM+PS is trying to obtain a value of 0.0894 and does poorly compared to the LSTM at 0.0934 ± 0.0015 . The authors should also include some idea of their proportional error -- e.g., they obtain a 2% error in III1.

A few small questions/comments:

- I cannot find sufficient training and validation details in the SI. How did the authors avoid overfitting their data? For which data points are errors reported in the text?
- Conceptually, how do the authors reconcile that multiple P_{Γ^B} can correspond to s_j^B ?

What would happen if s_j^B maps to two unrelated (and potentially non-overlapping) P_Γ^B ?

- Figure 2c is unreadable.

- Is it possible to use separate letters for s_j and $s^((t))$? It might be good to include a small section on notation at the beginning of the theory section.

Sincerely,
Rose K. Cersonsky

Reviewer #3 (Remarks to the Author):

In this manuscript, the authors propose a way to implement dynamical constraints into recurrent neural networks (RNNs) with inspiration from trajectory sampling methods. In particular, they introduce a cost function with a solution which exponentially biases trajectories with respect to some observables (constraints) in a way such that the average is some fixed value, provided by existing theory. They apply their methods to a variety of interesting problems, highlighting how their method is superior to other approaches involving RNNs.

The manuscript is generally well presented, well explaining their approach and their results. Their figures well illustrate and help the reader understand the methods. However, it is unclear to me how maximising Eq.3 gives Eq.4. Perhaps this could use a little explanation. Nevertheless, the method has a wide avenue of application. In particular, I see this being a very useful tool in the field of large deviations and rare event sampling.

I found the application to open quantum systems very interesting. The authors are able to use data from the results of one dissipation parameter to very well predict the results of another. This is surprising to me, as the data is generated using a quantum-jump monte carlo algorithm with an effective Hamiltonian which depends on this dissipation parameter (in addition to the stochastic nature also depending on the parameter). The evolution under this effective Hamiltonian is deterministic, and so I would expect the two parameters not to match up. This could be a consequence of the fact the dynamics is mostly dominated by the classical stochastic components. This difficulty of capturing all the quantum mechanic effects is well explained by the authors however.

I also think it is important to note that conditioning quantum jump trajectories to have observable averages equal to some other parameter does not give the exact same dynamics as simply running the dynamics with the different parameter. Indeed, one can do a "thermodynamics of quantum jump trajectories" (see J.P. Garrahan, I. Lesanovsky, PRL 104, 160601 (2010)) and define an s -ensemble for open quantum dynamics... The fact one cannot easily relate this ensemble to another with a different parameter makes it very difficult to calculate. I believe the approach presented here could provide a competitive way to sample the s -ensemble, an area which currently has few methods of doing so.

A few final questions which might be of interest:

- 1) It is my understanding the examples here are all done in discrete time (for OQS, discretizing time is often a necessary approximation for trajectory sampling). How easy is it to generalize this to continuous-time classical stochastic trajectories?
- 2) The focus here is conditioning using the s -ensemble to have a fixed average value. In this sense, it's a soft constraint where only the average is constrained. Is it possible to propose a cost function which can deal with hard constraints, where the observables of trajectories must take a certain value?

Response to reviewer #2

We thank the reviewer for the encouraging report and for the excellent suggestions for further improvement. We have implemented the suggestions in the revised manuscript. In what follows, we have reproduced all the comments of this reviewer in italics, followed by our response in bold after asterisk. Any changes resulting due to the comments of this reviewer are highlighted in color red in the revised manuscript.

REVIEWER COMMENTS

Reviewer #2 (Remarks to the Author):

The authors have proposed a novel application and instantiation for long short-term memory neural networks to mirror path-dependent constraints. The work is interesting, well-done, and novel; however, the writing and presentation of the paper make it somewhat difficult to follow and the authors should revise the article to make it more approachable for the likely general audience of Nature Communications.

In revisions, the authors can strengthen this paper by contextualizing the need for including path constraints in the LSTM model in the introduction (rather than focusing on the recent accomplishments of NNs), clarifying their procedure, and restructuring their figures to present their results more clearly (I have included some suggestions below).

According to the text, the authors train an LSTM to predict a path probability P_{Γ^A} , which they then use to subselect a set of paths P_{Γ^B} , using the principle of Maximum Caliber such that equation (6) corresponds to the target observable s_j^B . From here, they train a new LSTM to reconstruct their P_{Γ^B} , from which they can obtain the target observable at longer time scales.

Firstly, I'm hesitant to call this a variant of an LSTM (i.e., using a new name of ps-LSTM) or "incorporating" these constraints into the NN, as, in my reading, the authors do not change the actual infrastructure or mathematics of the LSTM procedure, but rather use the Maximum Caliber principle to do data sub-selection. For example, it's not necessarily fair to compare results obtained from an unconstrained and a constrained LSTM -- they are both using the same LSTM procedure, but with different training data. As it is more reflective of the procedure, the authors should rename "ps-LSTM" to "LSTM+PS" to denote the combination of the two methods, rather than an augmentation of LSTM.

*** Note that we didn't propose our method as a new variant of LSTM. "ps-LSTM" is only used to label the results and avoid confusion with the LSTM results. At this point we very respectfully request the reviewer to let us use the label ps-LSTM or path sampling-LSTM. This reflects that reweighting of paths and their proper sampling is fundamental to the protocol.**

Comparison with unconstrained LSTM is still important since the original training data is generated from the unconstrained LSTM. Here the LSTM model is used as a generative model and can give stochastic results, so there is still a need to show that the constrained version of LSTM is different from the unconstrained one.

Concerning sub-selection, what is the typical percentage of paths retained? One could argue that the number of simulations needed to approximate P_{Γ^A} should be pre-emptively increased considering the reduction to the subset – how did the authors ensure that their training sizes were sufficient in each LSTM model? Note that this is a different question from asking whether the training sizes were sufficient to distinguish P_{Γ^A} and P_{Γ^B} , as the authors cover this nicely in SI VI.

*** This is an excellent question. From our experience, the typical percentage of paths needed depends on what we want to constrain and how much the target value deviates from the unconstrained one. For example, in order to constrain the number of transitions, that means we need to have enough transitions in the sampled subset. The farther away the target value is from the unconstrained one, we would need to have a larger number of simulations. Therefore, we usually make a histogram of s_j for the paths generated from the unconstrained LSTM. We also plot Eq. (6) vs different $\Delta\lambda$ and compare it with the target value. If the target value doesn't cross the curve, then that means the subset size is not large enough to constrain the LSTM to the target value.**

However, we should admit that this way does not guarantee any theoretical support. The actual number of sufficient samples can therefore be a good future work. In sec III.2.1 we provide typical numbers used here: “We then perform path sampling with a biased probability $\propto e^{-\Delta\lambda}$ to select 10 trajectories from the 200 predictions. These 10 time series were then used to construct a subset and train a new ps-LSTM.” In sec III.2.2, “In order to constrain $\langle N \rangle$, we generate 800 independent time series from the constraint-free LSTM, and sample a subset consisting of 10 time series.”

Given that the training procedure includes two approximations: 1) approximating a subset of paths along the target path probability and 2) using this subset to extrapolate longer trajectories, where do the authors attribute their sources of error? It would be good to include an estimate of s_j^B for the intermediary subset in the main text to understand where the subset sits with respect to the initial and target path distributions.

*** First of all, the errors can come from the sampling of the simulations. Secondly, as can be expected and partially reported in the SI, the selections of subsets can also produce errors. We would argue that the extrapolation errors come from the training of LSTM and the quality of the subset, so as the reviewer suggested we have provided the estimated s_j^B of subsets we used in the SI in Supplementary Note VI.**

The authors should restructure their figures to report the errors with respect to the target value rather than the unconstrained LSTM procedure (as discussed above). For example, in Figure 2,

LSTM+PS achieves a value of 0.1271 ± 0.0001 with a target of 0.13; however, in the current table, it could appear that the LSTM+PS is trying to obtain a value of 0.0894 and does poorly compared to the LSTM at 0.0934 ± 0.0015 . The authors should also include some idea of their proportional error – e.g., they obtain a 2% error in III1.

*** This is a good point. While we have left the concise Fig. 2 in the main text (except Fig 2c as suggested by the reviewer in another point below), in the SI in Supplementary Fig. 1a we have now provided error estimates from the target/theory values for various observables.**

A few small questions/comments:

- I cannot find sufficient training and validation details in the SI. How did the authors avoid overfitting their data? For which data points are errors reported in the text?

*** We trained our LSTM model using the same procedure in our previous work (Nat. Commu. 11, 5115 (2020), Ref. 26), in which we chose an appropriate number of training epochs such that the training loss is starting to be greater than validation loss. Regarding the second question here, the figure captions for different systems provide details of the error calculation for the free energies and transition rates, for instance: “For reference MD, the error bars were calculated by averaging over transition time in a single 100ns or 200ns trajectory, while for the predictions from LSTM and ps-LSTM, the error bars were averaged over 10 independent predictions with the transition time for each predicted trajectory calculated in the same way as MD trajectories.”**

- Conceptually, how do the authors reconcile that multiple P_{Γ^A} can correspond to s_j^B ? What would happen if s_j^B maps to two unrelated (and potentially non-overlapping) P_{Γ^A} ?

*** This is a good question where we need to delineate path probability from path. The same path probability can lead to different paths fluctuating around it. LSTM, as we showed in our original 2020 Nat Commun and also here, can stochastically sample the ensemble of paths such that their average will have the right values of different observables. With the above preamble, the answer to this question then lies in Eq 5 for tilting of path ensembles. For a given target s_j and value of λ , Eq 5 shows a unique ensemble of paths should exist. However, if there are 2 or more possible values of λ itself, then there are 2 or more ensembles of paths with the same desired observable. This would be intriguing if true and is worthy of further investigation. However in our numerical tests for fitting λ we do not see such a scenario.**

- Figure 2c is unreadable.

We have now moved the original Figure 2c to the SI where it is much more expanded, and have also provided the associated numbers there.

- Is it possible to use separate letters for s_j and $s^{\{t\}}$? It might be good to include a small section on notation at the beginning of the theory section.

*** We thank the reviewer’s suggestion and have changed the one-hot vectors $s^{\{t\}}$ to $v^{\{t\}}$.**

Sincerely,
Rose K. Cersonsky

Response to reviewer #3

We thank the reviewer for the very encouraging report. In what follows, we have reproduced all the comments of this reviewer in italics, followed by our response in bold after asterisk. We have also implemented the suggestions in the revised manuscript. Any changes resulting due to the comments of this reviewer are highlighted in color blue in the revised manuscript.

Reviewer #3 (Remarks to the Author):

In this manuscript, the authors propose a way to implement dynamical constraints into recurrent neural networks (RNNs) with inspiration from trajectory sampling methods. In particular, they introduce a cost function with a solution which exponentially biases trajectories with respect to some observables (constraints) in a way such that the average is some fixed value, provided by existing theory. They apply their methods to a variety of interesting problems, highlighting how their method is superior to other approaches involving RNNs.

The manuscript is generally well presented, well explaining their approach and their results. Their figures well illustrate and help the reader understand the methods. However, it is unclear to me how maximising Eq.3 gives Eq.4. Perhaps this could use a little explanation. Nevertheless, the method has a wide avenue of application. In particular, I see this being a very useful tool in the field of large deviations and rare event sampling.

*** We have added a reference (Z. Faidon Braozakis, M. Vendruscolo, and P. G. Bolhuis, PNAS 118(2), e2012423118 (2021)) which has an elegant explanation using Maximum Caliber after Eq. 4, which we think is better for helping readers outside the field understand the derivation without unnecessary equations and paragraphs.**

I found the application to open quantum systems very interesting. The authors are able to use data from the results of one dissipation parameter to very well predict the results of another. This is surprising to me, as the data is generated using a quantum-jump monte carlo algorithm with an effective Hamiltonian which depends on this dissipation parameter (in addition to the stochastic nature also depending on the parameter). The evolution under this effective Hamiltonian is deterministic, and so I would expect the two parameters not to match up. This could be a consequence of the fact the dynamics is mostly dominated by the classical stochastic components. This difficulty of capturing all the quantum mechanic effects is well explained by the authors however.

I also think it is important to note that conditioning quantum jump trajectories to have observable averages equal to some other parameter does not give the exact same dynamics as simply running the dynamics with the different parameter. Indeed, one can do a "thermodynamics of quantum jump trajectories" (see J.P. Garrahan, I. Lesanovsky, PRL 104, 160601 (2010)) and

define an s-ensemble for open quantum dynamics... The fact one cannot easily relate this ensemble to another with a different parameter makes it very difficult to calculate. I believe the approach presented here could provide a competitive way to sample the s-ensemble, an area which currently has few methods of doing so.

We thank the reviewer for the kind comments.

A few final questions which might be of interest:

1) It is my understanding the examples here are all done in discrete time (for OQS, discretizing time is often a necessary approximation for trajectory sampling). How easy is it to generalize this to continuous-time classical stochastic trajectories?

*** The reviewer is correct that the trajectory we used to train our LSTM must be of discrete time. Although there does exist continuous-time RNN/LSTM, we are not sure whether it is critical for learning stochastic time series data such as MD or quantum jump trajectories. Whether or not it is possible to extend our approach to continuous-time classical stochastic trajectories depends on the learning performance of continuous-time RNN/LSTM but not really on the path sampling approach we proposed, so we would say as long as continuous-time RNN/LSTM learns well on continuous-time stochastic trajectories, it should be easy to write down similar constraint and select the subset. Though due to the computational limitations, the constraint will then need to be something computable and discretized. We will explore this interesting avenue in the future!**

2) The focus here is conditioning using the s-ensemble to have a fixed average value. In this sense, it's a soft constraint where only the average is constrained. Is it possible to propose a cost function which can deal with hard constraints, where the observables of trajectories must take a certain value?

*** This is an excellent question. Since we use our LSTM to generate paths of any length, it might not make sense to use a hard constraint. However, in general, one might decide to fix the length of paths they want to generate then use a hard constraint.**

REVIEWERS' COMMENTS

Reviewer #2 (Remarks to the Author):

The authors have addressed the majority of both my and Reviewer 3's comments; however, there are still some outstanding comments from the initial review that have not been fully addressed.

- In revisions, the authors can strengthen this paper by contextualizing the need for including path constraints in the LSTM model in the introduction (rather than focusing on the recent accomplishments of NNs), clarifying their procedure, and restructuring their figures to present their results more clearly (I have included some suggestions below).

The authors have provided no further context for the importance of introducing path constraints. This will provide better motivation for the study and the necessary context for the very general audience of Nature Communications.

- The authors should also restructure their figures to report the errors with respect to the target value rather than the unconstrained LSTM procedure (as discussed above). For example, in Figure 2, LSTM+PS achieves a value of 0.1271 ± 0.0001 with a target of 0.13; however, in the current table, it could appear that the LSTM+PS is trying to obtain a value of 0.0894 and does poorly compared to the LSTM at 0.0934 ± 0.0015 .

The authors have only minorly amended Figure 2 and have not made any further amendments to clarify their figures. Many figures are still difficult to read.

Once the authors have made these changes, I would be happy to recommend acceptance.

Sincerely,
Rose K. Cersonsky

Reviewer #3 (Remarks to the Author):

The authors have satisfactorily addressed the minor points and questions I asked.

Response to reviewer #2

We thank the reviewer for further pointing out a few of her outstanding suggestions. In what follows, we have reproduced all the comments of this reviewer in italics, followed by our point by point response in bold after asterisk. Any changes resulting due to the comments of this reviewer are highlighted in color red in the revised manuscript.

REVIEWERS' COMMENTS

Reviewer #2 (Remarks to the Author):

The authors have addressed the majority of both my and Reviewer 3's comments; however, there are still some outstanding comments from the initial review that have not been fully addressed.

• In revisions, the authors can strengthen this paper by contextualizing the need for including path constraints in the LSTM model in the introduction (rather than focusing on the recent accomplishments of NNs), clarifying their procedure, and restructuring their figures to present their results more clearly (I have included some suggestions below).

*** We thank the reviewer for the suggestion of emphasizing the importance for adding path constraints. In the second paragraph of Introduction, we have provided a context to emphasize such importance through a common situation in the MD community which has also been studied using our second example (Aib9+kappa) later:**

“For instance, in the context of generalizing and extrapolating time-series observed from finite length simulations or experiments, partial sampling when generating a training dataset is almost unavoidable. This may come from only being able to simulate dynamics on a particular timescale that is not long enough to completely capture characteristics of interest or simply due to thermal noise. This could then manifest as a misleading violation of detailed balance, and a RNN model trained on such a time-series would dutifully replicate these violations. In such cases, enforcing physics inspired constraints corresponding to the characteristics of interest when training an RNN-based model is critical for accurately modeling the true underlying distribution of data.”

Adding kinetic or path constraints could also be important in other applications, however, as we are not experts in those areas, we decided to focus on their approach instead of talking about the importance of adding path constraints in their fields.

The authors have provided no further context for the importance of introducing path constraints. This will provide better motivation for the study and the necessary context for the very general audience of Nature Communications.

*** Please see our reply above.**

• The authors should also restructure their figures to report the errors with respect to the target value rather than the unconstrained LSTM procedure (as discussed above).

*** We have moved the plot of each individual probability to Supplementary Fig. 2 b and expanded it to page wide width to increase the visibility. In Fig. 2 a we have also provided a table listing each value plotted in Fig. 2 b and provided each error percentage between ps-LSTM probability and its targeted theoretical calculation. In Fig. 2 c, we have now added a new column reporting the error compared with the target constraint value instead of LSTM value.**

For example, in Figure 2, LSTM+PS achieves a value of 0.1271 ± 0.0001 with a target of 0.13; however, in the current table, it could appear that the LSTM+PS is trying to obtain a value of 0.0894 and does poorly compared to the LSTM at 0.0934 ± 0.0015 .

*** In the Supplementary Information we have now provided the kappa values of the subset which equal around 0.1348. In other words, the statistics of kappa is clearly different from the original training dataset which gives a kappa value of 0.0894. Considering that ps-LSTM is no different from the LSTM except it learns from a different training dataset sub-selected from the original dataset with the new kappa value, it is very unlikely that it will try to learn the original kappa value in the original dataset. It is more likely that it does learn perfectly from the subset with 0.1348 kappa and only reaches 0.1271, thus close within error bars.**

The authors have only minorly amended Figure 2 and have not made any further amendments to clarify their figures. Many figures are still difficult to read.

*** We thank the reviewer for the suggestion. The plot of each individual probability has been moved to SI and expanded to page wide width to increase visibility. We have also provided a table for listing and comparing the value of probabilities with their theoretical predictions.**

For Fig. 3 and 4, however, we decided to keep the original figures because we not only care about their predictions of constraints but also their statistics and kinetics. Showing only error percentages of constraints isn't sufficient to demonstrate how well ps-LSTM performed. On the other hand, it is difficult to obtain theoretical predictions of statistics and kinetics such as free energy profiles and the transition times. As a result, we compare them with the statistics and kinetics of the MD trajectories. The corresponding errors are then calculated using multiple ps-LSTM predictions and filled with transparent color.

Once the authors have made these changes, I would be happy to recommend acceptance.

Sincerely,
Rose K. Cersonsky